# Validation of a QTL on Chromosome 1DS Showing a Major Effect on Salt Tolerance in Winter Wheat

**DOI:** 10.3390/ijms232213745

**Published:** 2022-11-08

**Authors:** Maisa Mohamed, Md Nurealam Siddiqui, Benedict Chijioke Oyiga, Jens Léon, Agim Ballvora

**Affiliations:** 1INRES Plant Breeding, Rheinische Friedrich-Wilhelms-University, 53115 Bonn, Germany; 2Agronomy Department, College of Agriculture, South Valley University, Qena 83523, Egypt; 3Kleinwanzlebener Saatzucht (KWS) KWS SAAT SE & Co. KGaA, 37574 Einbeck, Germany

**Keywords:** biparental populations, gene expression, QTL, salt stress, wheat

## Abstract

Salt stress is one the most destructive abiotic stressors, causing yield losses in wheat worldwide. A prerequisite for improving salt tolerance is the identification of traits for screening genotypes and uncovering causative genes. Two populations of F_3_ lines developed from crosses between sensitive and tolerant parents were tested for salt tolerance at the seedling stage. Based on their response, the offspring were classified as salt sensitive and tolerant. Under saline conditions, tolerant genotypes showed lower Na^+^ and proline content but higher K^+^, higher chlorophyll content, higher K^+^/Na^+^ ratio, higher PSII activity levels, and higher photochemical efficiency, and were selected for further molecular analysis. Five stress responsive QTL identified in a previous study were validated in the populations. A QTL on the short arm of chromosome 1D showed large allelic effects in several salt tolerant related traits. An expression analysis of associated candidate genes showed that *TraesCS1D02G052200* and *TraesCS5B02G368800* had the highest expression in most tissues. Furthermore, qRT-PCR expression analysis revealed that *ZIP-7* had higher differential expressions under saline conditions compared to *KefC*, *AtABC8* and *6-SFT*. This study provides information on the genetic and molecular basis of salt tolerance that could be useful in development of salt-tolerant wheat varieties.

## 1. Introduction

Wheat is the third most important crop grown extensively worldwide, with global production and use now estimated at 775.6 and 755.8 million tons, respectively [1]. It contributes about 20% of total dietary calories and protein and plays a strategic role in global economy, industry, food and nutrition [2,3,4,5].

Despite the importance of wheat for global food security, its productivity has been severely reduced due to multiple abiotic stresses including salinity, drought, cold, ion toxicity, etc. [6]. Among these stressors, salinity in lower-rainfall environments is considered to be one of the most important environmental factors reducing crop productivity and indicative of global nutritional balance [6].

Wheat is moderately tolerant to soil salt concentrations, showing no significant yield losses at 6 dS m^−1^ [7,8] and a 50% yield loss at 13 dS m^−1^ [7]. Adverse effects of salinity on plant growth can be due to osmotic stress and ion cytotoxicity [9]. With an excess of Na^+^ ion accumulation and low water potential of soil, hyperosmotic and hyper-ionic stress occur, in addition to primary stresses [10]. These results are manifested in decreased germination percentage, reduced growth and yield, and changes in reproductive behavior [11].

As a result, the ability of plants to deal with salt stress is of great importance for maintaining wheat production [12]. Thus, it is necessary to convene all available tools of conventional and modern plant breeding for the development of salt-tolerant cultivars that can meet the increasing demand for wheat production [13,14].

Although there are several strategies to increase wheat production in salt-affected areas (such as leaching and drainage), cultivation of salt-tolerant genotypes is known to be the most effective way to increase wheat production in agroecological regions with soil salinity [15]. Crops that are tolerant at the seedling stage may also show improved salinity tolerance at the adult stage [15,16,17,18,19]. This means that when a high genetic correlation between early and late stage response exists, a selection for salt tolerance at the seedling stage can serve as a direct, rapid substitute for identifying plants with salt tolerance at the adult stage.

Through previous studies, phenotyping for salt tolerance at the seedling stage has been exploited successfully in the identification of salt tolerance genotypes [15,20,21] and detection of genetic mechanisms of salt tolerance in wheat [22,23,24]. It has revealed that shoot growth is more sensitive to salt stress than the root growth: firstly, the reduction in leaf area development relative to the root growth leads to a reduction in water use by the plant; secondly, the accumulation of Na^+^ and/or Cl at toxic concentration levels disturbs the photosynthetic capacity [25]. Previous reports have also shown that in segregated populations of wheat, genetic variations for salinity tolerance exist [26].

Salt stress has been found to alter the morphological, physiological and biochemical responses of plants [27]. Salinity causes various physiological disturbances resulting from osmotic stress, ion toxicity and imbalance of nutrient elements in the cytoplasm of plant cells [28,29]. Salinity affects chlorophyll (Chl) content in many crops by adversely affecting Chl synthesis or accelerating its degradation, which reduces photosynthetic capacity [30]. The plant’s ability to maintain Chl levels under salt stress is characterized in wheat as a salt resistance trait [31]. The production and accumulation of free amino acids, especially proline, by plant tissues under abiotic stress is considered an adaptive response [27]. Proline has been thought to perform like a compatible solute that regulates the osmotic potential in the cytoplasm [32,33]. Salt stress disturbs cytoplasmic K^+^/Na^+^ homeostasis, resulting in a reduction in the cytosolic K^+^/Na^+^ ratio [34]. The accumulation of excess Na^+^ and Cl^−^ causes an ionic imbalance that can damage the selectivity of root membranes and induce K^+^ deficiency in plants [35]. In wheat, genotypic variation in salt tolerance has been associated with low rates of Na^+^ transport and high selectivity for K^+^ over Na^+^ [36,37]. Therefore, it is important to assess whether genotypes with different salinity tolerance (ST) use ion exclusion as a tolerance mechanism against salinity [38]. In addition, salinity stress causes stomatal closure and hinders carbon dioxide (CO_2_) entrance in leaves. This restrains CO_2_ fixation and enables the chloroplast to produce immense levels of energy, which further develops the reactive oxygen species (ROS) [39,40,41,42,43,44]. These ROS cause damage to major molecules including lipid, protein, and nucleic acids [41,42]. ROS production increased under salinity [43] and induced cellular toxicity in various crop plants [39]. It has been observed that salt stress (100 mM NaCl) enhanced malondialdehyde (MDA) level up to 35% or 68% after 5 or 10 days of exposure to such stress, respectively, in wheat seedlings [45].

Water deficits affect a cascade of physical, signaling, gene expression, biochemical, and physiological pathways and processes, resulting in decreased cell elongation, wilting, and, ultimately, plant death; these harmful effects of salinity can be considered as water-deficit effects [46,47,48].

Salt tolerance is a quantitative trait for which numerous loci (QTL) have been reported in wheat at the stages of germination, seedling and maturity, and for plant survival [49].

Genetic variation for ST at different growth stages in wheat offer a great opportunity for ST improvement. Various genes are involved in the increase of plant tolerance to salt stress by regulating diverse mechanisms including the antioxidant defense system, Na^+^ exclusion, maintenance of Na^+^/K^+^ homeostasis, transpiration efficiency, and cytosolic K^+^ retention [50,51]. Several strategies are used to reduce yield losses under salt stress using conventional breeding tools [11]. Recent developments in molecular markers combined with the wheat reference sequence allow plant scientists to understand the genetic basis of complex phenotypic traits by using thousands of genetic markers in a genome-wide association study (GWAS), thereby identifying marker-trait associations (MTAs). Several reports have already shown that the application of high-density marker arrays for single nucleotide polymorphisms (SNPs) in combination with a GWAS can identify relevant MTAs and help postulate candidate genes underlying the complex phenotypic traits [52,53]. In the previous studies, several MTAs regulating ST in wheat have been identified using a broad panel of cultivars [22,23]. Therefore, this study was performed to validate the candidate genes underlying the QTLs and show how segregation of these QTLs affects ST levels among their progenies by testing the segregation in two independent bi-parental populations. Thus, this research will enhance our understanding of holistic salinity tolerance mechanisms and will aid in the breeding of salt-tolerant wheat lines.In detail, the present study was conducted to achieve the following objectives:(i)To characterize the salinity tolerance of F_3_ lines of two connected biparental crosses, in which a salt-sensitive parent (Bobur) was crossed with two different salt-tolerant cultivars (Altay2000) and lines (UZ-11CWA08), respectively.(ii)To describe the ionomic, biochemical and physiological responses of salt-sensitive and salt-tolerant lines.(iii)To validate the effect of salt-tolerant candidate genes identified in the prequel wheat study.(iv)To analyze the gene expression of putative candidate genes that is involved in the salinity response of the lines. The results should be valuable for improving salt tolerance and breeding salt-tolerant cultivars faster.

## 2. Results

### 2.1. Phenotypic Traits Were Affected by Salinity Stress

Three lines that showed consistent responses to salinity across at all growth stages tested were selected as parents for crossing [15]. Bobur, the salt-sensitive parent, was crossed with Altay2000 and UZ-11CWA08, the salt tolerant parents [15]. The F_1_ progenies of these crosses were selfed and from the resulting F_2_ plants 274 and 277, respectively, F_3_ lines were established by further selfing in the Bobur*Altay2000 and Bobur*UZ-11CWA08 cross. These lines (F_3_ generation) were tested under two salt treatments under hydroponic conditions for shoot and root traits.

The populations were significantly different from each other in the tested traits and showed significant interactions with the salt treatments (Appendix A), which justifies a population-wise analysis. The ANOVAs showed that the salinity treatment resulted in a significant reduction in trait scores in both populations. For instance, a reduction of 61.82% in SFW, 18.31% in SDW, 28.57% in RFW, and 6.41% in RDW were observed in Bobur*Altay2000 population, while the Bobur*UZ-11CWA08 population showed a reduction of 51.53% in SFW, 16.21% in SDW, 35.71% in RFW and 10.52% in RDW (Table 1). In both populations with the exception of RDW for the genotype effects, there were highly significant genotype and genotype by salinity interaction effects in all traits tested.

In both populations, the genotype values for the traits under control and salt stress conditions were normally distributed (Appendix A). The coefficients of variation (CV) for all traits were lower under salinity compared to salinity-free treatment in both populations and parents, with the exception of parents UZ-11CWA08 and Bobur (for SFW and SDW), for which the coefficients of variation (CV) were higher under saline conditions (Appendix A).

The H² was calculated on a population basis and showed higher values for the shoot traits than for the root traits. Comparing fresh and dry weight H^2^ values, the fresh weight values were always higher (Appendix A). Perusal of results on heritability revealed low to moderate heritability estimates for all the measured traits.

### 2.2. Identifying Salt-Tolerant and Salt-Sensitive Progenies in the Segregating Populations

In order to classify the progenies of the populations into salt-tolerant and salt-sensitive lines, several indices were calculated from the relationships between the stress and control treatments. To identify the index that describes the shoot water loss (SWL) [15], the correlation between the respective index and the SWL was calculated (Appendix A). TOL and SSI showed highly significant correlations with SWL, whereas STI was highly significantly negatively correlated to SWL. The biplot plot showed that the SSI and TOL indices were clustered together with SWL, while the STI index was opposite to them. The MP and GMP clustered together but give different information compared to SWL (Appendix A for Bobur*Altay2000 and Appendix A for Bobur*UZ-11CWA08).

MP and GMP were neither highly correlated with SWL nor with each other. Consequently, STI, TOL, and SSI were the most appropriate indices among all evaluated indices, including SWL, for ranking the progeny.

Using the selected indices simultaneously including SWL and based on the entire ST rank list (Appendix A), lines 52, 84, 83 and 51 were categorized as tolerant, moderately tolerant, moderately sensitive and sensitive to salt stress for the 274 F_3_ lines of the cross Bobur*Altay2000 and lines 49, 84, 84 and 45 were categorized as tolerant, moderately tolerant, moderately sensitive and sensitive to salt stress for 277 F_3_ lines of the cross Bobur*UZ-11CWA08. Mean ST estimates ranged from 13.14 and 10.97 in salt-tolerant lines to 84.28 and 83.64 in salt-sensitive lines for lines 274 and 277 F_3_ of the Bobur*Altay2000 and Bobur*UZ-11CWA08 cross; the overall means were 49.53 and 48.55, respectively (Appendix A).

The ranking results showed that the contrasting lines were four (salt-tolerant lines) versus two (salt-sensitive lines) and two (salt-tolerant lines) versus two (salt-sensitive lines) for F_3_ lines of cross Bobur*Altay2000 and F_3_ lines of cross Bobur*UZ-11CWA08, respectively. These lines showed a consistent response to salt stress in five replications under a hydroponic system. These lines were P1G082, P1G119, P1G202 and P1G264 (salt-tolerant) andP1G132 and P1G253 (salt-sensitive) for F_3_ lines of the Bobur*Altay2000 cross, while lines were P2G076 and P2G243 (salt-tolerant) and P2G027 and P2G162 (salt-sensitive) in the contrasting F_3_ lines of the Bobur*UZ-11CWA08 cross (Appendix A).

### 2.3. Differences between Salt-Tolerant and Salt-Sensitive Lines

#### 2.3.1. Salt Stress Response of Leaf Ionic Traits in Salt Contrasting Lines

In order to observe the influence of salt stress in the contrasted lines on different ion accumulations, Na^+^ and K^+^ contents were determined in leaves of wheat lines under control and salt stress. Significant differences in ion accumulation were evident in wheat lines after exposing to salt stress as compared to control conditions (Figure 1, Appendix A), The application of salt stress led to a significant increase in Na^+^, as well as decrease in K^+^ contents of leaf in wheat lines (Figure 1). We found that leaf Na^+^ content varied depending on the line. Bobur has higher leaf Na^+^ contents (10.18) than Altay2000 (5.61) and UZ-11CWA08 (4.93) (Figure 1A). Analysis of the Na^+^ and K^+^ contents of the selected lines revealed that the salt-tolerant lines have higher K^+^ and lower Na^+^ content when compared with the salt-sensitive lines (Figure 1B,C,E,F). For instance, in F_3_ lines of cross Bobur*Altay2000, Na^+^ content increased in the leaf with increasing salinity, and higher contents were noted in P1G132 and P1G253 (salt-sensitive) (9.1 and 9.9, respectively); lesser Na^+^ contents were noted in P1G082, P1G119, P1G202 and P1G264 (salt-tolerant) (4.76, 8.63, 4.86 and 7.70, respectively) (Figure 1B). In F_3_ lines of Bobur*UZ-11CWA08, higher contents of leaf Na^+^ were noted in P2G027 and P2G162 (salt-sensitive) (4.63 and 5.20, respectively) while lesser Na^+^ contents were noted in P2G076 and P2G243 (salt-tolerant) (4.53 and 4.12, respectively) (Figure 1C).

A reduction in cellular K^+^ contents in leaves of wheat lines was recorded with increasing levels of salt stress while K^+^ content was higher in the salt-tolerant lines than in the salt-sensitive lines under salt stress (Figure 1D–F). We found that, initially, K^+^ contents in leaves decreased with increase in salinity; reductions in salt-tolerant parents Altay2000 and UZ-11CWA08 were 38.45% and 41.23% respectively. The magnitude of this reduction in the salt-sensitive parent Bobur was 50.91% compared to the plants under control conditions (Figure 1D). Lower K^+^ contents were found in P1G132(20.82) and P1G253(32.17) (salt-sensitive) while higher K^+^ contents were noted in P1G082(36.95), P1G119(50.59), P1G202(36.13) and P1G264 (48.81) (salt-tolerant) (Figure 1E). Lines P2G027 (46.85) and P2G162 (47.72) (salt-sensitive) had lower K^+^ contents than lines P2G076 (49.01) and P2G243 (60.9) (Figure 1F).

The K^+^/Na^+^ ratio was significantly affected by salinity. Increasing the Na^+^ contents led to a decrease in the K^+^/Na^+^ ratio in all wheat lines (Appendix A). However, salt-tolerant lines, including the salt-tolerant parents, showed a minimum reduction in the K^+^/Na^+^ ratio compared to the sensitive ones (Appendix A).

#### 2.3.2. Biochemical and Physiological Modulations under Salt Stress

Proline is considered to be among a very effective class of compatible solutes and a wide range of crop plants have been reported to accumulate it under abiotic stress such as salinity and drought [54]. Biochemical analysis of leaves of different wheat lines for proline accumulation showed that proline accumulation increased significantly under saline conditions (Figure 2, Appendix A). Compared to the salt-tolerant lines, the salt-sensitive lines showed a higher accumulation of proline content (Figure 2). In the parents, the highest percentage increase relative to the level under control conditions was observed in Bobur, in which proline concentration increased by approximately 312.14%. The lowest increase was observed in Altay2000, in which it increased by 203.24% under salt stress (Figure 2A).

Regarding the F_3_ lines of Bobur*Altay2000, the highest and lowest percentage increases in proline accumulation relative to that under control conditions were found in P1G253 (the salt-sensitive line (112.73%)) and P1G119 (the salt-tolerant line (4.48%), respectively (Figure 2B).

In F_3_ lines of Bobur*UZ-11CWA08, P2G162 (the salt-sensitive line) had the highest percentage increase proline accumulation relative to that under control conditions (927%); the lowest increase observed in P2G243 (the salt-tolerant line) was 16.83% (Figure 2C).

Interestingly, we found that the pattern of proline accumulation in the parents and both populations shows the same behaviour in salt-tolerant lines and salt-sensitive lines (Figure 2).

To analyze the salt induced physiological changes in the contrasting lines in the segregating populations and their parents, chlorophyll (Chl) content and chlorophyll fluorescence were measured in these lines after exposure to salt treatment.

It had been found that the percentage reduction in total Chl content was more at the higher salinity levels compared to control and the data obtained from the measurements show that the Chl content was consistently higher in the control treatment than in the saline treatment (Appendix A), The Chl content of the leaves decreased with salinity (Figure 3A–C). The results showed that the Chl content of salt-tolerant lines decreased slightly under salt conditions (Altay2000 (28.57%), UZ-11CWA08 (36%) and P1G082 (23.52%), P1G119 (27.11%), P1G202 (22.41%), P1G264 (28.30%) and P2G076 (18.75%), P2G243 (15.09%)). However, the maximum decrease was in salt-sensitive lines (Bobur (58.20%) and P1G132 (43.10%), P1G253 (47.69%) and P2G027 (48.52%), P2G162 (55.26%) compared to tolerant lines (Figure 3A–C).

The FluorPen FP100 for chlorophyll-a fluorescence detection (OJIP assay) is a highly sensitive technique for evaluating PSII photochemistry in addition to electron transport efficiency and has been extensively used to study the integrity and activity of the photosynthetic apparatus [55]. The effects of salinity on the shape of the chlorophyll fluorescence transition curve are shown in Appendix A. Salt stress significantly inhibited fluorescence transients across all OJIP phases. The transient fluorescence curve of the lines showed a slight decrease in J and I steps compared to the control group. Calculated parameters from chlorophyll-a fluorescence are presented in Appendix A. Under salt stress, Fv/Fm, Fo/Fm, and Fv/Fo decreased in salt-tolerant lines and increased in salt-sensitive lines (Appendix A). The fixed area estimates decreased in all lines under salt stress, but the decrease was much greater in salt-sensitive lines than in salt-tolerant lines (Appendix A).

#### 2.3.3. Validation of Candidate Genes in Both Segregating Populations and by Expression Analysis

To validate the QTLs identified in a GWAS [22], they were analyzed in the contrasting lines. The first step was to examine whether the marker alleles of the QTL regions could distinguish these contrasting lines. For that more than 40 SNP markers of the QTL regions were selected and tested in the parents and offspring (Table 2).

Several marker alleles of the progenies showed exactly the same allelic pattern of classification as the phenotypic classification of the contrasting lines. In the Bobur*Altay2000 population, we found a separation by alleles between the lines on Chr. 1DS (position 108.87 cM) and Chr. 2BS (position 367.4 cM), which corresponds to the behavior of the parents. In population Bobur*UZ-11CWA08, we also found the QTL on Chr. 1DS (position 108.87 cM) and additionally a QTL region on Chr. 5BL (position 280.68 cM) to be informative. Several other markers, however, showed a segregation pattern of their alleles, which does not correspond to the phenotypic classification. Although we found segregating markers for most QTL regions, not all of them segregated in both populations. Five of the markers were polymorphic in both populations (Table 2).

The contrasting lines were analyzed by an ANOVA using the respective alleles as a factor. The markers on Chr. 1DS (position 108.87cM) showed significant allelic effects for SFW, SDW, RFW, sodium content and proline accumulation in the Bobur*Altay2000 population and for SFW, SDW, RFW, potassium content, proline accumulation and chlorophyll content in the Bobur*UZ-11CWA08 population. For the same QTL region, ANOVA revealed significant markers by salinity treatment interaction effects for SFW, SDW, sodium content, potassium content, proline accumulation, and Fixed-Area in the Bobur*Altay2000 population and for SFW, SDW, RFW, proline accumulation, and chlorophyll content in the Bobur*UZ-11CWA08 population.

While both populations possessed these effects for the QTL region on 1DS, exactly the same marker which showed the same pattern of allelic segregation in the Bobur*Altay2000 on QTL 2BS (position 367.4 cM) did not show a comparable classification into the salt-sensitive or salt-tolerant group in the Bobur*UZ-11CWA08 population and did not reveal significant allelic effects for the tested traits. On the other hand, the significant QTL of Chr. 5BL (position 280.68 cM) in the Bobur*UZ-11CWA08 population could not be found in the Bobur*Altay2000 population.

Consequently, these QTLs (Table 3) were used to detect the informative genes and proteins involved in the response to salinity stress and (major) cell regulatory mechanisms. The TGACv1 genome sequence assembly version of *Triticum aestivum* L. publicly available on JBrowse, was used to detect these genes (Appendix A).

Using the WheatGmap web tool and based on these QTLs (Table 3), we found a wide range of expression for the candidate genes in different cereal tissues and at different developmental stages (Appendix A). Among the candidate genes, *TraesCS1D02G052200* in 1DS and *TraesCS5B02G368800* in 5BL showed the highest expression in most organs and tissues, indicating that they play important roles during development, growth and grain filling. *TraesCS5B02G368500* in 5BL showed semi-highest expression in first leaf blade, flag leaf and leaf. *TraesCS1D02G052700* in 1DS was highly expressed in the endosperm. Four of the significantly associated genes, *TraesCS1D02G054400*, *TraesCS1D02G054500* and *TraesCS1D02G054600* which were on 1DS, and *TraesCS2B02G503100* on 2BS, showed very low expression in the tissues compared to the other genes.

The candidate genes which were validated in the parents [22] were further validated in the contrasting lines for both segregating populations. Quantitative Real-Time PCR (qRT-PCR) was conducted to quantify the kinetics of *ZIP7* (zinc transporter), *KefC* (glutathione-regulated potassium-efflux system protein), *AtABC8* (ABC transporter B family member 8) and *6-SFT* (sucrose: fructan-6-fructosyltransferase) expression in parents and contrasting progenies under control and salt stress conditions. Figure 4 and Figure 5 show the relative expression of *ZIP7*, *KefC*, *AtABC8* and *6-SFT* at 42 DAS (day after saline) for the contrasting parents and progenies, calculated according to the algorithm described in [56]. At day 42, the expression of *ZIP7*, *KeFc*, *AtABC8* and *6-SFT* showed that they are upregulated in salt-tolerant lines, including parents (Altay2000, UZ-11CWA08) and lines (P1G082, P1G119, P1G202 and P1G264) and (P2G076 and P2G243) (Figure 4 and Figure 5). In contrast, the salt-sensitive lines including parent (Bobur) and some lines (P1G132 and P1G253) and (P2G027 and P2G162) were down–regulated (Figure 4 and Figure 5). The *ZIP-7* exhibited higher differential expression compared to the *KefC*, *AtABC8* and *6-SFT* expression in both, contrasting parents and progenies (Figure 4 and Figure 5).

## 3. Discussion

The biology of cellular plants strengthens our understanding of the diverse network of traits related to salinity tolerance and increases the structural genomics and functional methods suitable for use in the detection of the quantitative trait loci (QTLs) genes of interest linked with specific traits [57]. Therefore, understanding salt tolerance mechanisms and analyzing salt stress-related genes and their functions will provide a theoretical basis for understanding the stress signal network and pathways for the improvement of the target crop [58]. Whereas plants’ tolerance mechanism is a complex phenomenon and depends upon physiological and genetic responses [59]. They indicated that these processes involve phenotypic evaluation as well as the identification of QTLs closely related to molecular markers. Identification of new QTLs may lead to the development of new salt-tolerant lines [59].

Improving salt tolerance is a great challenge as it is a very complex trait that is under polygenic control [60]. To improve wheat production in salinity-affected areas, access to adequate genetic diversity is critical for current and future breeding efforts. Considerable effort has been made to identify salt-tolerant wheat genotypes. In the present study, a wide range of phenotypic variability was observed for all tested traits between F_3_ offspring from two crosses between salt-sensitive and salt-tolerant parents. An analysis of genetic differences was performed to verify proposed [22] marker-trait associations showing salt-tolerant genomic regions. Verified QTL regions and a postulation of candidate genes can accelerate breeding for new high-yielding genotypes that are also salt-tolerant.

The heritabilities of the shoot traits were higher than those of the root traits. However, before a selection of the lines with regard to their shoot traits can be recommended, the correlations of the shoot traits to the target traits must be considered. Whereas in both populations, TOL and SSI for SFW and SDW showed highly significant correlations with SWL, however STI was highly significant negatively correlated to SWL. Also, MP and GMP for SFW and SDW were neither highly correlated with SWL nor with each other.

In general, lines under salt stress show higher Na^+^ levels. Here, salt-sensitive lines showed the highest Na^+^ concentration in the leaf compared to the salt-tolerant lines (Figure 1A–C). The magnitude of this reduction in the salt-sensitive lines were higher in Bobur (312.14%), P1G132(209.52%), P1G253(285.21%) P2G027 (110.45%) and P2G162 (131.11%) compared with salt-tolerant lines which were Altay2000 (204.89%), UZ-11CAW08 (285.15%), P1G082 (110.61%), P1G119 (175.71%), P1G202(82.02%) P1G264 (270.1%), P2G076 (93.58%) and P2G243 (67.47%). A small or zero increase in Na^+^ under stress indicated that these genotypes were more tolerant than those that translocated high levels of Na^+^ into their leaves. Higher concentrations of Na^+^ impede various metabolic activities [61]. The different accumulations of Na^+^ of the various lines show that there are genetic differences in this trait and that genotypes that accumulate only low amounts of Na^+^ in their leaves react genetically differently to the increased salinity [62]. An uptake or transport mechanism that distinguishes similar ions such as Na^+^ and K^+^ could be a useful selection criterion for salt tolerance in wheat and breeding for efficient nutrient uptake [63].

With increased hydroponic salt concentration, all wheat lines showed reduced K^+^ levels (Figure 1D–F). The decrease in K^+^ levels is due to the presence of excess Na^+^ in the growing medium, as high external Na^+^ levels are known to have an antagonistic effect on K^+^ uptake in the plant [64]. Salt tolerance is definitely related to K^+^ content, since K^+^ is involved in osmotic regulation and there is competition between these two ions [65]. Salt stress drastically affects growth and decreases the process of photosynthesis due to the imbalance of the internal ionic concentration of the various cation and anions like sodium and potassium [66].

The lines show genetic differences in the K^+^/Na^+^ ratio, with the salt-tolerant lines having significantly higher K^+^/Na^+^ ratios compared to the salt-sensitive lines (Appendix A). As with the data on K^+^/Na^+^ ratios indicate that Na^+^ exclusion from leaf tissues plays a critical role in rice salt tolerance by maintaining the optimal K^+^/Na^+^ ratio [35].

It has been found that stressful environments such as drought, salinity and high temperatures can cause damage to the plant at any stage [67]. In addition to molecular, physiological and chemical damage, there is primarily direct damage in the photosynthesis process, which can impair the growth of plants at every growing stage and thus reduce the yield [67].

Proline concentration in the leaves increased significantly under salinity (Appendix A), with the highest increased by approximately 312% (Bobur), 112.73% (P1G253) and 927% (P2G162) in salt-sensitive lines when compared with the salt-tolerant lines (Figure 2). This finding is consistent with the observation of who also observed that salt-sensitive cultivars increased proline levels under salt stress [68]. Contrary to these reports, exogenous proline treatment often results in higher salt tolerance [69]. The present results suggest that increasing proline concentration may not be associated with salt tolerance, consistent with similar observations previously reported by [70] were made. Proline accumulation did not seem to perform a part in the improved salt tolerance of the amphiploid relative to that of wheat (Chinese Spring). The levels of Proline were higher in leaf blades of the salt-sensitive wheat (Chinese Spring) than in those of the more salt-tolerant amphiploid (except in the oldest leaf blade) [70].

However, elevated proline levels may also confer additional regulatory or osmoprotective functions under salt stress, such as its role in the control of the activity of plasma membrane transporters involved in cell osmotic adjustment in barley roots [71]. Given the fact that proline biosynthesis is a highly energy-demanding process and that only small quantities of proline are probably required for the control of plasma membrane transporters [71], the observed over production of proline in sensitive genotypes may not be explained by these processes, but rather may reflect of poor performance and greater damage in response to salt stress. Consequently, selecting for higher proline levels to increase salt tolerance can be counterproductive.

More studies are required on more morphological and physiological traits to understand the complexity of salt tolerance [72,73,74,75]. A reduction in the chlorophyll content was observed in all lines under salt stress. This reduction was more pronounced in the salt-sensitive lines than in the salt-tolerant ones (Figure 3) (Appendix A) may be due to the replacement of Mg^2+^ with Na^+^ in these sensitive genotypes [76,77]. Across different plants exposed to salinity stress, the reduction in chlorophyll content is an indicative response [78]. In soybean seedlings were also observed a significant reduction in chlorophyll content at high NaCl level [79]. The reduction in chlorophyll content caused a reduction in photosynthesis [60]. The salt-tolerant wheat genotypes revealed higher levels of chlorophyll content compared to the salt sensitive group [60]. Thus, the chlorophyll content would be useful in screening large numbers of genotypes [80].

The chlorophyll fluorescence transients (Fo, Fj, Fi, Fm and Fv) in contrasting parents and both salt-tolerant and salt-sensitive lines declined (Appendix A) under saline conditions. The decrease in Fo due to salt stress indicates an increased thermal dissipation [81,82], while the decrease in Fv may be attributed to the pigment losses due to salt injury [15]. Salinity stress reduces photosynthesis by inhibiting photosystem II complex (PSII) at both acceptor [QA] and donor side (oxygen evolving complex OEC) and destruction of chlorophyll pigments by accumulation of toxic ions [83]. It has been suggested the use of fluorescence induction parameters to detect metabolic perturbations by abiotic stresses [84]. Under salt stress, the Fv/Fm, Fm/Fo and Fv/Fo declined in salt-tolerant lines and increased in the salt-sensitive lines (Appendix A), suggesting different mechanisms are controlling these physiological traits in wheat, making them useful parameters for distinguishing salt—tolerant from salt-sensitive genotypes [15]. The fixed area was higher in the salt-tolerant lines compared to the salt-sensitive ones (Appendix A). The result of our study was consistent with the findings in [15] which reported that salt stress had negative effect on the fixed area in wheat genotypes.

For the development and improvement of salt-tolerant cultivars, it is important to identify the relevant genes that determine and/or influence salt tolerance in plants. The aim of this study was to validate putative candidate genes controlling salt tolerance in wheat by validating QTL regions and using reference sequences to identify them. In both tested biparental populations we found a strong allele effect from members of the QTL on Chr. 1DS on 108.87cM. Due to a co-segregation of further tested QTL positions, it might be that a QTL on 2BS in the cross Bobur*Altay2000 or of a QTL on 5BL in the cross Bobur*UZ-11CWA08 are also responsible for salt tolerance. However, in both of these latter positions we could not find any phenotypic response in the respective other population of the biparental crosses with the common parent Bobur. Therefore, we conclude that the responsive allele is located on the QTL region of chr. 1DS at position 108.87cM.

The literature findings of the putative candidate genes can be described as follows:

By using the WheatGmap web tool, the candidate genes, *TraesCS1D02G052200* in 1DS and *TraesCS5B02G368800* in 5BL showed the highest expression in most organs and tissues of wheat, indicating that they play important roles during development, growth and grain filling (Appendix A), whereas, *TraesCS1D02G052200* encodes alcohol dehydrogenase (ADH). ADH had made some variation in glycolysis and alcohol fermentation, and under flooding stress enhanced the germination of transgenic soybeans [85]. *AT1G64710* and *AT5G24760,* which are considered other ADH family genes, were reactive in the root and leaf in Arabidopsis working alongside during the PEG-induced water stress, supporting the conclusion that in response to drought, the capacity for ethanolic fermentation was improved [86]. In Arabidopsis, another ADH gene conferring both biotic and abiotic stress resistance has been reported in [87]. By regulating the ROS-related genes to maintain the ROS homeostasis in sugarcane, the *ScADH3* gene as one of the ADH genes appeared to affect cold tolerance [88].

Interestingly, *TraesCS5B02G368800* encoding cation-chloride cotransporters (CCC) was detected on chromosome 5B and was found to be highly expressed in wheat leaves (fifth leaf sheath and fifth leaf blade) (Appendix A). This gene is important because in yeast to maintain fluid/ion homeostasis, during osmotic and oxidative stress, CCC (e.g., Na^+^/K^+^/2Cl^−^ cotransporters (NKCC) and K-Cl cotransporters) are activated [89]. The NKCC plays a vital role in osmotic regulation and cell ionic adjustments, as an integral membrane protein transporting Na^+^, K^+^ and 2Cl^−^ [90,91].

*TraesCS1D02G052700* in 1DS was highly expressed in the endosperm (Appendix A). It encodes the sugar transporter SWEET gene, a member of the glycoprotein SWEET gene family which performs a vital role in plant growth and development, and in response to environmental stress. As reported in [92], during the early phases of salt stress, Kentucky bluegrass with SWEET genes exhibited a rise in their expression compared with controls.

In osmotic stress tolerance, SWEETs may also play an important role [93]. For example, during senescence and osmotic stresses including cold, high salinity and osmotic stress, *AtSWEET15*, also known as *SAG29* (senescence-associated gene 29) is strongly induced [94]. Comparable to *OsSWEET5* [95], under normal growth conditions, constitutive overexpression of *AtSWEET15* consequences in enhanced leaf senescence. Compared to control plants, *AtSWEET15* overexpression lines show reduced root growth and cell viability under high salinity conditions [93]. Root growth decreasing in *atsweet15* mutant lines is like to that of the control in contrast [93]. However, compared to the control and over expression lines root cells are more viable in the mutant [94]. So, this is a suggestion that during osmotic stress *AtSWEET15* may perform a role in modulating cell viability [93]. It has been revealed that SWEET proteins play crucial roles in plant development and stress responses and in the plant kingdom this is considered one of the largest sugar transporter families [96]. In tea plants, CsSWEET genes play important roles in the response to abiotic and biotic stresses and offer insights into the characteristics of SWEET genes, which could serve as the basis for further functional identification of such genes [97].

*TraesCS5B02G368500* in 5BL showed semi-highest expression in first leaf blade, flag leaf and leaf (Appendix A), encoding a potassium transporter. This indicated that the gene *ApKUP4* (alligator weed K^+^ transporter gene) which contributes to salinity tolerance in transgenic Arabidopsis seedlings, is essential for plant salinity tolerance and potassium homeostasis [98]. The results reported in [99] demonstrate that, in rice, *OsHAK1* (a high-affinity potassium transporter, which positively adjusts responses to drought stress) is considered a drought-responsive gene; its expression is related to increased dehydration tolerance through the systemic regulation of K^+^ homeostasis, proline accumulation, root system architecture, plasma membrane protection, and stimulation of stress-related genes. It is important to improve abiotic stress tolerance in cereals at the seedling and reproductive stages of plants grown under osmotic and water-limiting conditions: *OsHAK1* gene overexpression does not cause any growth defect indicating that overexpression of this ion transporter gene is a hopeful approach [99]. In the presence of toxic concentrations of Na^+^, the high-affinity potassium transporter HKT (High-affinity K^+^ transporters) gene family can selectively uptake K^+^ in halophytic plants [100]. SbHKT_1;4_ expression was upregulated more strongly in a salt-tolerant sorghum accession, correlating with a better balanced Na^+^/K^+^ ratio and improved plant growth, upon Na^+^ stress [100].

Regarding the qRT-PCR results of *ZIP7* [101], *KefC* [102], *AtABC8* [103] and *6-SFT* [104] genes showed specific expression patterns in shoots of salt-tolerant (up-regulation) and salt-sensitive (down-regulation) lines indicating that they are involved in ST (Figure 4 and Figure 5). Further analyses of the transcription of these genes in the root cells are essential, as the organ that is in close contact with the solution.

## 4. Materials and Methods

### 4.1. Plant Materials

In a previous study, one hundred and fifty wheat accessions were characterized for their root and shoot water loss due to salt stress and it was found that several accessions possessed higher ST potentials [15]. From this panel, three genotypes that showed consistent responses to salinity at all growth stages tested were chosen as parents for the cross between salt-sensitive and salt-tolerant combinations. Bobur, classified as salt-sensitive, was crossed with Altay2000 and UZ-11CWA08, both classified as salt-tolerant [15]. These two biparental populations were selfed twice to establish F_2_ derived F_3_-lines (F_2:3_) (Appendix A). Together with their parents, the harvested F_3_ lines (274 lines from 1st population Bobur*Altay2000 cross and 277 lines from 2nd population Bobur*UZ-11CWA08 cross) were then used for ST evaluation in a hydroponic system at seedling stage. The contrasting lines for both segregating populations were selected by using ST rank consisting of the selected indices, STI (stress tolerance index), SSI (stress susceptibility index), TOL (tolerance index) and SWL (shoot water loss).

### 4.2. Hydroponic Experiment

The screening was performed at seedling stage under hydroponic systems using the modified Hoagland solution as described by [105]. Two independent experiments, Experiment 1 (E1) and Experiment 2 (E2), using the two populations were conducted in the greenhouse (of Crop Science and Resource conservation Institute (INRES), University of Bonn, Germany) following randomized complete block design (RCBD) with five replications (five hydroponic experiments) for each line. In E1 (including 274 segregating F_3_ lines of cross Bobur*Altay2000) (August–November 2019) and E2 (including 277 segregating F_3_ lines of cross Bobur*UZ-11CWA08) (December–March 2020), the lines were screened with non-saline (control) and saline (115 mM NaCl) nutrient solution; supplementary Ca^2+^ as CaCl_2_ was added to the saline nutrient solution in 20:1 molar ratio of NaCl [106], to improve nutrient uptake and ameliorate the effects of salinity on the plant. In each replication, comparisons were made between saline and non-saline conditions for each line of both populations. The electrical conductivity (EC) values for control and saline (115 mM NaCl) (+5.0 mM CaCl_2_) solutions ranged as follows for the first and second F_3_ line of (Bobur*Altay2000 and Bobur*UZ-11CWA08): 1.79–1.84, 14.24–15.44 and 1.79–1.84, 14.24–15.44 dS m^−1^, respectively. The supported hydroponic system is followed by [15].

### 4.3. Trait Quantification

35 days after planting (DAP), shoot and root were separated and weighed to obtain fresh shoot weight (SFW) and fresh root weight (RFW). The harvested samples (separated shoots and roots) were oven dried at 60 °C for 4 days and weighed to obtain the shoot dry weight (SDW) and root dry weight (RDW).

The relative water content (RWC) of shoot and root was calculated using the method proposed by [107] on the basis of FW and DW in stress conditions (S) vis-a-vis the control conditions (C): WL = [(FWC − DWC) − (FWS − DWS)]. (WL, water loss; FWC, fresh weight under control; DWC, dry weight under control; FWS, fresh weight under stress; DWS, dry weight under stress.)

### 4.4. Ionic, Biochemical and Physiological Analyses

To estimate Na^+^ and K^+^ content, leaf samples of the contrasting lines were oven dried at 65 °C for 3 days and ground for determination of mineral composition. Ash from leaf samples was dissolved in 5.1% HNO_3_ and used to determine the Na^+^ and K^+^ contents with an atomic absorption spectrophotometer (Type 2380; Perkin Elmer, Wellesley, MA, USA), and then the K^+^/Na^+^ ratios were calculated.

Proline was measured from leaf samples, according to [108], adapted to a microplate-based protocol [109].

Plant chlorophyll content index (CCi) of the leaves in terms of SPAD (soil plant analysis development) values was measured using a portable SPAD-502 meter (Minolta, Osaka, Japan). The leaf chlorophyll content index was measured from the leaf tip to the leaf base of each line and then averaged. The age of the highly salt-tolerant and salt-sensitive wheat lines was 6 weeks under saline and non-saline conditions.

Chlorophyll-a fluorescence (ChlF) of leaf samples from 6-week-old highly salt-tolerant and salt-sensitive wheat lines under saline and non-saline conditions was measured with the FluorPen FP100 (Photon Systems Instruments, Brno, Czech Republic). The OJIP parameters were calculated as in [15]. The OJIP parameters were analyzed using PSII efficiencies (Fv/Fm = quantum yield of PSII, Fo/Fm = non-photochemical loss in PSII and Fv/Fo = efficiency of the water − splitting complex on the donor side of PSII).

### 4.5. Analysis of Marker Trait Associations (MTAs) and Their Allelic Variations

40 MTAs for salinity tolerance have been identified [22]. In this study, we aimed to validate the segregation of these MTAs between groups of contrasting offspring by designing primers around the associated SNP (single nucleotide polymorphism) using the online program “primer3” (http://primer3.wi.mit.edu/, accessed on 29 October 2021). DNeasy Plant Mini Kit (Qiagen, Hilden, Germany) was used to extract DNA from dried plant tissues stored at room temperature. Standard polymerase chain reaction (PCR) protocol was followed for amplification of DNA. Thus, 100 ng of DNA template in 25 μL of 1× One Taq Standard Buffer (Biolabs, Ipswich, MA, USA), 0.2 mM dNTPs and 0.2 μM of each primer were amplified with 0.5 units of One Taq DNA polymerase (Biolabs, Ipswich, MA, USA). Cycling conditions were established with an initial denaturation step at 95 °C/2 min followed by 40 cycles at 95 °C/45 s, annealing temperature (Ta) for 45 s, extension at 72 °C/1 min per kbp, and a final extension step at 72 °C/5 min.

Locus-specific sequence primers were designed according to the method described in [110]. The amplified fragments were purified using the QIAquick PCR purification kit (Cat.No:-FG-91302). Sanger sequencing was performed by Eurofins Genomics (Ebersberg, Germany).

### 4.6. In Silico Expression Analysis of Candidate Genes

The expression profiles of all putative candidate genes associated with the identified SNPs were taken from the published wheat RNA-seq expression database on the WheatGmap web tool (https://www.wheatgmap.org, accessed on 5 March 2022 [92]).

### 4.7. RNA Extraction and qRT-PCR Analyses

Total RNA was extracted from the harvested leaf samples after 42 days under saline and non-saline conditions using the RNeasy Plant Mini Kit (Cat# 74903 and 74904). cDNA synthesis was performed using the LunaScript RT SuperMix Kit (NEB #E3010) (New England BioLabs, Ipswich, MA, USA). A qPCR was performed using the Luna Universal qPCR Master Mix Kit (NEB #M3003) (New England BioLabs). Gene expression data were analyzed using the standard methods of [56], normalized with two internal control genes, TaEf-1a and TaEf-1b (Unigene accession: Ta659). The design of the gene primers and the sequences of the qRT-PCR primers are described in [22].

### 4.8. Statistical Analyses

Before statistical analysis, STI (Stress Tolerance Index), SSI (Stress Susceptibility Index), TOL (Tolerance Index), MP (Mean Productivity), and GMP (Geometric Mean Productivity) were calculated using the formulas described in [111].

The lines were ranked for each salt tolerance index and the overall ST ranking for each line was calculated as follows:$$\mathrm{ST}\;\mathrm{overall}=\sum_{Z}^{M}ST\;rankings$$
where *Z* is the index of the ST estimates of lines for each measured traits and *M* is the number of measured traits across replications. Lines with extreme response to salt stress were identified as follows: tolerant (ST < 25th percentile) and sensitive (STg > 75th percentile).

For analysis, firstly, an adjusted best linear unbiased estimator (BLUE) was calculated for each entry for all the different traits to correct for errors due to planting positions (row-and-column effects) in the hydroponic tubes by including “Replication/Row*Column” which means that rows crossed with columns were nested within replication [112]. The adjusted phenotypic values were analyzed population-wise and thereafter combined for both populations in ANOVAs using PROC GLM (SAS version 9.4) according to the following models:Population-wise: *Y_iklr_* = *μ* + *S_i_* + *g_k_* + *b_l_*(*S*_i_) *+ e_ijlr_*
Combined: Y_ijklr_ = *μ* + *S_i_* + *P_j_* + *S***P_ij_* + *g_k_*(*P_j_*) + *b_l_*(*S_i_*) + *e_ijklr_*
where *Y_ijklr_* is the adjusted phenotype (trait value) of the *k*th genotype of the *j*th population grown in the *i*th salt treatment in the *l*th block in the hydroponic system; *μ* is the general mean, *S_i_* is the fixed effect of the *i*th salt treatment, *P_j_* is the fixed effect of the *j*th population, *S***P_jk_* is the fixed effect of the *j*th population grown under *i*th salt treatment (interaction), *g_k_*(*P_j_*) is the random effect of the *k*th genotype of the (nested in) population *P_j_*, *b_l_*(*S_i_*) is the random effect of the *l*th block within *i*th salt treatment and *e_ijklr_* represents the error term. Fixed effects are denoted as uppercase letters, random effects are denoted by lowercase letters, interaction is indicated by “*”, and nesting is indicated by “()”. The Duncan multiple range test was used to compare the genotypic mean values.

The variance components due to genotypic (σ^2^_g_) and error (σ^2^_e_) effects for each treatment were estimated on basis of the adjusted BLUE values using the REML option in PROC VARCOMP [113].

Furthermore, we calculated broad-sense heritability H^2^, for each hydroponic treatment and population by applying REML option in Proc varcomp:H^2^ = σ^2^_g_/[σ^2^_g_ + (σ^2^_e_/r)],
where σ^2^_g_ is the genotypic variance, σ^2^_e_ is the residual error variance, and r is the number of replications. The heritability was categorized as low, moderate and high as given by [114].

Pearson correlation analysis of genotypic means was performed to assess the correlation between RWC and other quantitative indices of salt tolerance using the package *Performance Analytics* and the PCA for some salt tolerance indices and SWL was performed by *Factominer* and *Factoextra*, both also implemented in (R software version 3.6.3 [115]).

Histograms of the measured traits were analyzed to find the distribution of the measured traits by using (R software version 3.6.3 [115]).

## 5. Conclusions

Stress tolerance indices were used to classify F_3_ lines of two crosses into salt-tolerant and salt-sensitive lines with respect to their response to salt stress. The identified contrasting groups showed markedly differential physiological, biochemical, and ionic responses to salt stress. The salt-tolerant lines from the 1st population (P1G082, P1G119, P1G202, P1G264) and 2nd population (P2G076, P2G243) of the cross Bobur*Altay2000 and Bobur*UZ-11CWA08, respectively showed a higher leaf K^+^/Na^+^ ratio, lower proline accumulation, higher chlorophyll content and higher rates of PSII photochemical activities. This study provides useful information regarding the phenotypic, ionic, biochemical and physiological variations found in the germplasm of the contrasting F_3_ wheat lines. Those traits, such as Na^+^ and K^+^ content, chlorophyll content, chlorophyll-a fluorescence (ChlF) and proline accumulation that effectively differentiate the salt-tolerant and salt-sensitive groups can be used for an indirect selection of salt tolerance. Further research is needed to verify the function and involvement of the candidate genes identified in this study and can then accelerate breeding improvement for wheat salt tolerance.

## Figures and Tables

**Figure 1 ijms-23-13745-f001:**
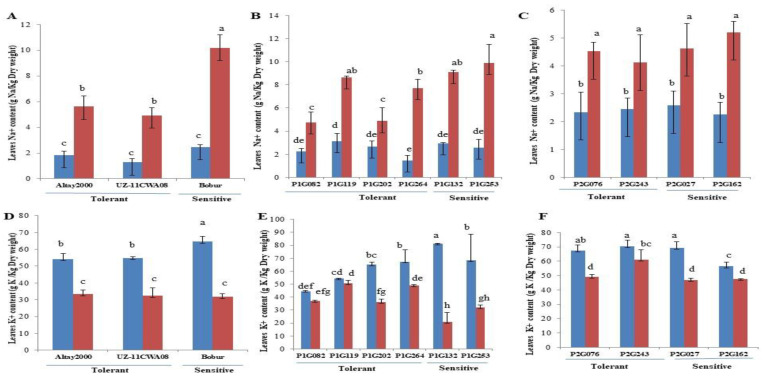
Effect of salinity on leaf Na^+^ and K^+^ content. (**A**,**D**) Parents. (**B**,**E**) Contrasting F_3_ lines of cross Bobur*Altay2000, (**C**,**F**) of cross Bobur*UZ-11CWA08. Non-saline (blue) and saline (red) conditions. Error bars represent SEs. Different letters denote statistically significant differences among the genotypes at *p* < 0.05 level detected by Duncan multiple range test.

**Figure 2 ijms-23-13745-f002:**
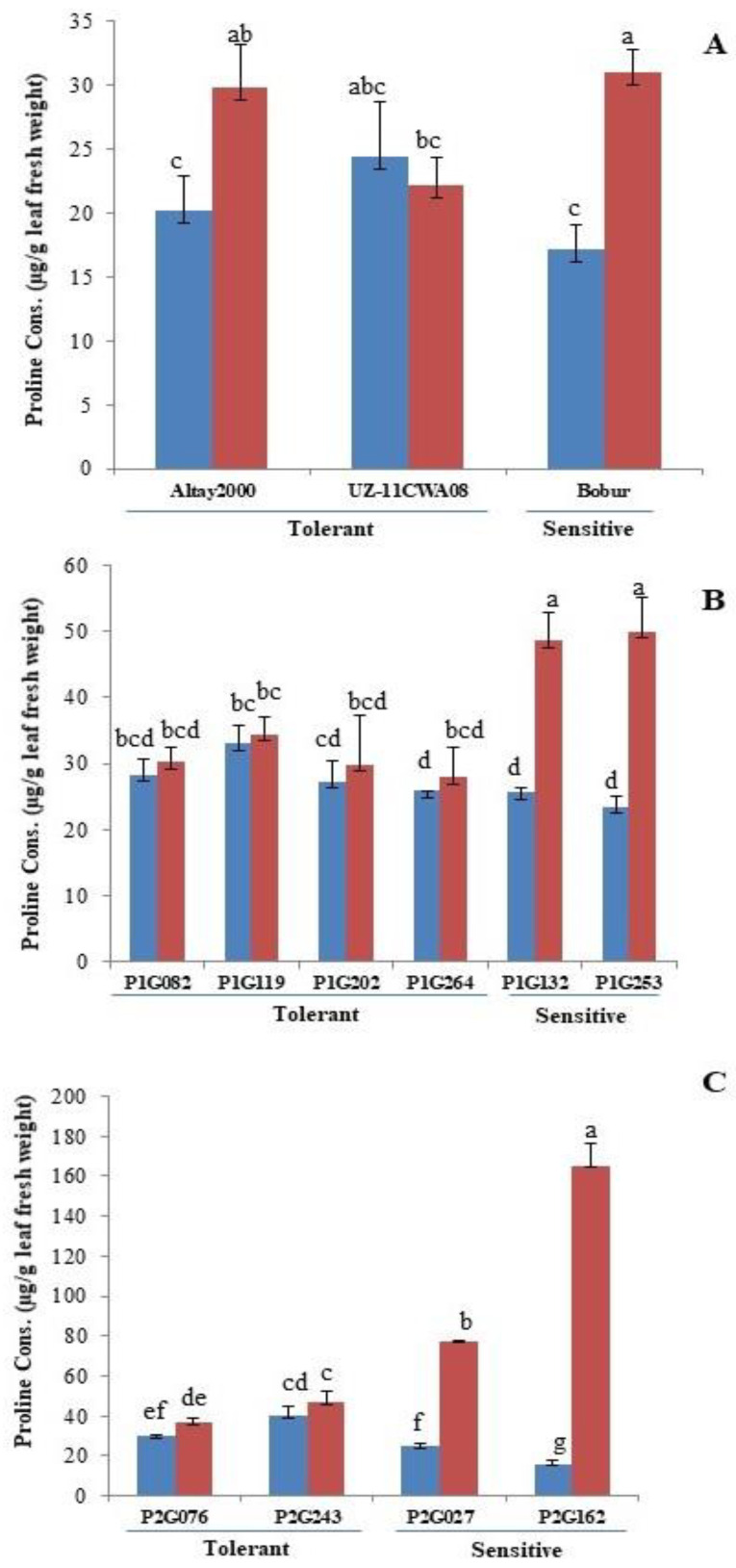
Effects of salinity on proline content. Parents (**A**). Contrasting F_3_ lines of cross Bobur*Altay2000 (**B**) and of cross Bobur*UZ-11CWA08 (**C**). Non-saline (blue) and saline (red) conditions. Error bars represent SEs. Different letters denote statistically significant differences among the genotypes at *p* < 0.05 level detected by Duncan multiple range test.

**Figure 3 ijms-23-13745-f003:**
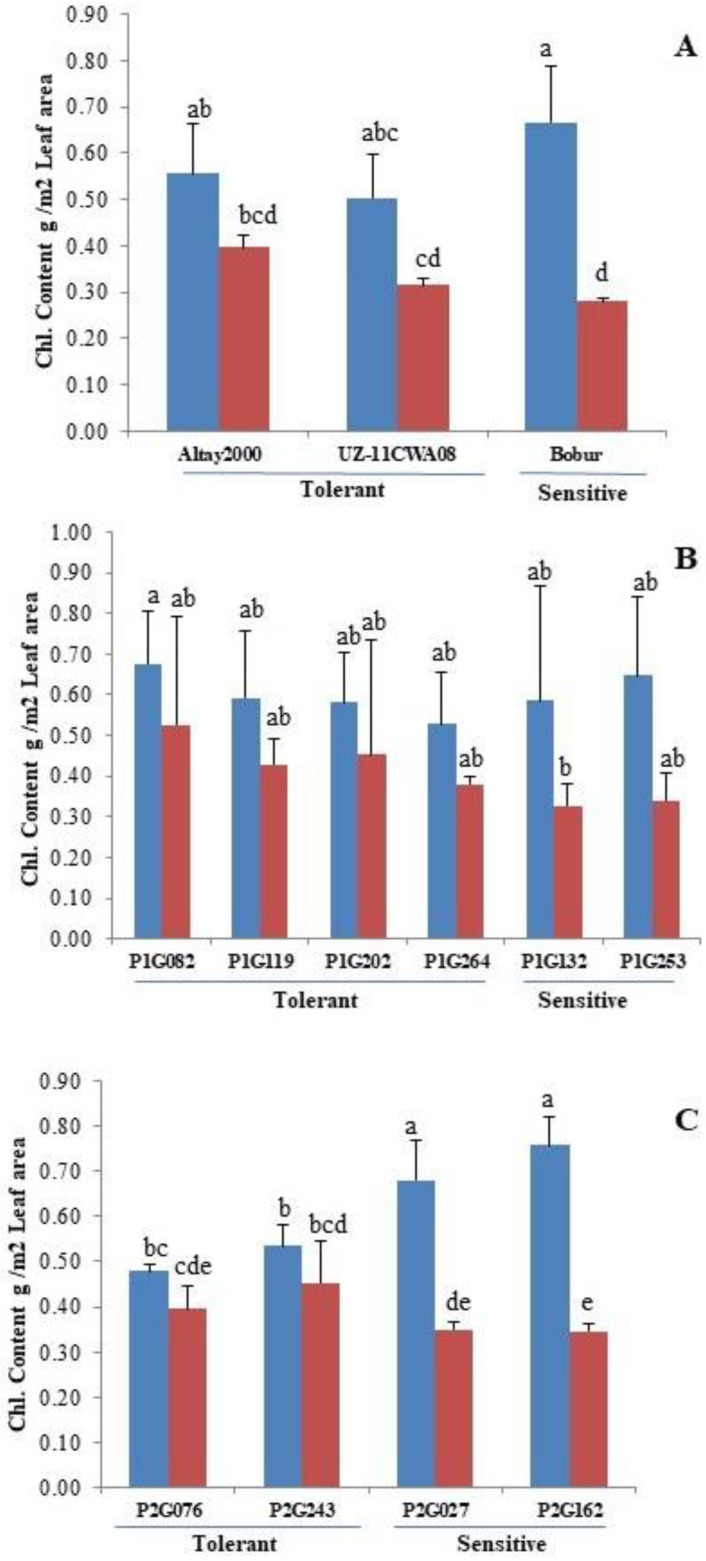
Effects of salinity on chlorophyll content. Parents (**A**). Contrasting F_3_ lines of cross Bobur*Altay2000 (**B**) and of cross Bobur*UZ-11CWA08 (**C**). Non-saline (blue) and saline (red) conditions. Error bars represent SEs. Different letters denote statistically significant differences among the genotypes at *p* < 0.05 level detected by Duncan multiple range test.

**Figure 4 ijms-23-13745-f004:**
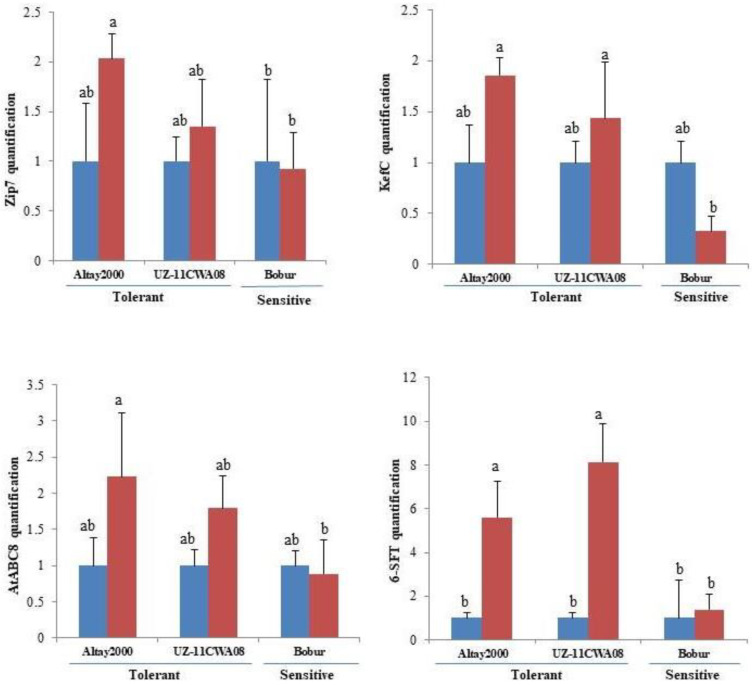
Expression levels of zinc transporter (*ZIP7*), glutathione-regulated potassium-efflux system protein (*KefC*), ABC transporter B family member 8 (*AtABC8*) and sucrose: fructan-6-fructosyltransferase (*6-SFT*) in leaves of two salt-tolerant (Altay2000 and UZ-11CWA-8) and salt-sensitive (Bobur) lines after 42 d in non-saline (blue) and saline (red) conditions, determined by 2-ΔCT method. Efa1.1 and Efa1.2 genes were used as internal control genes. Error bars represent SEs. Different letters denote statistically significant differences among the genotypes at *p* < 0.05 level detected by Duncan multiple range test.

**Figure 5 ijms-23-13745-f005:**
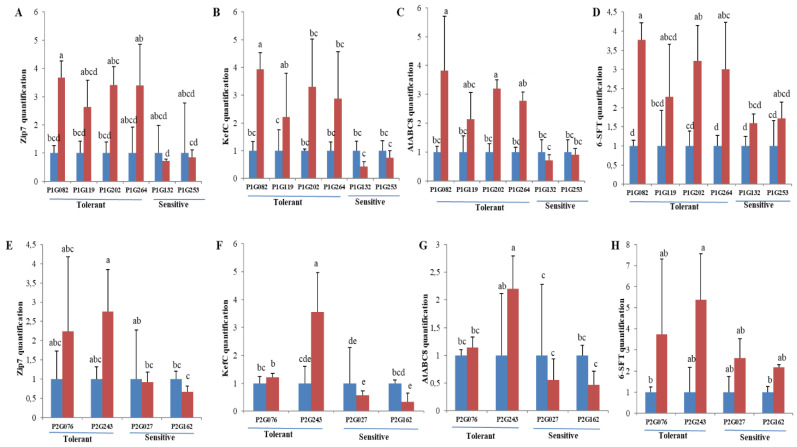
Expression levels of zinc transporter (*ZIP7*), glutathione-regulated potassium-efflux system protein (*KefC*), ABC transporter B family member 8 (*AtABC8*) and sucrose: fructan-6-fructosyltransferase (*6-SFT*) in leaves of (**A**–**D**) salt-tolerant (P1G082,P1G119,P1G202 and P1G264) and salt-sensitive (P1G132 and P1G253) contrasting F_3_ lines of cross Bobur*Altay2000, (**E**–**H**) salt-tolerant (P2G076 and P2G243) and salt-sensitive (P2G027 and P2G162) contrasting F_3_ lines of cross Bobur*UZ-11CWA08 after 42 d in non-saline (blue) and saline (red) conditions, determined by 2-ΔCT method. Efa1.1 and Efa1.2 genes were used as internal control genes. Error bars represent SEs. Different letters denote statistically significant differences among the genotypes at *p* < 0.05 level detected by Duncan multiple range test.

**Table 1 ijms-23-13745-t001:** F-tests and *p* values of main effects of salinity and genotypes and their interactions for the measured traits at seedling stage.

S.O.V	d.f	SFW	SDW	RFW	RDW
	F_3_ lines of cross Bobur*Altay2000				
Treatments (T)	1	12,343.8 ***	2172.64 ***	8817.58 ***	2117.06 ***
Genotypes (G)	273	8.01 ***	2.57 ***	6.60 ***	1.36 ^ns^
TxG	271	6.10 ***	1.94 ***	10.10 ***	1.68 ***
Error (Mean sq)	548	1.67	0.03	0.19	0.007
	F_3_ lines of cross Bobur*UZ-11CWA08				
Treatments (T)	1	11,679.6 ***	5825.92 ***	1440.62 ***	1915.45 ***
Genotypes (G)	276	7.88 ***	2.29 ***	1.68 ***	1.09 ^ns^
TxG	272	12.07 ***	9.86 ***	1.71 ***	3.64 ***
Error (Mean sq)	554	1.22	0.02	0.75	0.02

Note: F-values are shown; significance levels *p*: *** *p* ≤ 0.001; ^ns^, not significant; d.f, degree of freedom; SFW, shoot fresh weight; SDW, shoot dry weight; RFW, root fresh weight; RDW, root dry weight.

**Table 2 ijms-23-13745-t002:** SNP markers of the QTL regions in the parents and contrasting lines of both segregating populations.

SNP	Chr	Genome	Position (CM)	Alleles	Parents		Contrasting F_3_ Lines of Cross Bobur*Altay2000					
							P1G082	P1G119	P1G202	P1G264	P1G132	P1G253
					Altay2000	Bobur	Tolerant	Tolerant	Tolerant	Tolerant	Sensitive	Sensitive
RAC875_c38018_278	2AL	A	110.13	T/C	T	C	C	C	T	C	C	C
Kukri_c11327_977	2AL	A	341.14	T/G	G	T	T	T	T	T	T	T
Excalibur_c20439_825	2AL	A	497.75	T/C	C	T	T	T	T	T	T	T
Excalibur_c39151_104	2AL	A	502.19	A/G	A	G	G	G	G	G	G	G
BS00066475_51	3AL	A	275.6	A/G	G	A	A	A	A	A	A	A
RAC875_c16405_84	4AS	A	147.89	T/C	C	T	T	T	T	T	T	T
Tdurum_contig33628_129	4AS	A	147.89	T/C	C	T	T	T	T	T	T	T
tplb0024k14_1812	6AS	A	115.71	T/C	C	T	T	T	T	T	T	T
BS00035083_51	7AL	A	103.7	T/C	T	C	C	C	C	C	C	C
wsnp_Ex_c43009_49439922	7AL	A	103.7	T/C	T	C	C	C	C	C	C	C
Kukri_c1831_1243	7AL	A	150.81	T/C	T	C	C	C	C	C	C	C
Ex_c2725_1442	1BS	B	201.12	A/G	G	A	G	G	G	G	G	G
BobWhite_c11044_322	1BS	B	266.71	T/C	T	C	C	C	C	C	C	C
BobWhite_c43917_288	1BS	B	269.73	A/G	G	A	G	G	G	G	G	G
RAC875_c11609_62	2BS	B	277.23	A/G	G	A	A	A	A	A	A	A
**Ex_c16948_754**	**2BS**	**B**	**367.4**	**A/G**	**A**	**G**	**A**	**A**	**A**	**A**	**G**	**G**
BobWhite_c5756_532	2BS	B	583.38	A/C	A	C	C	C	C	C	C	C
Kukri_c54078_114	5BL	B	257.76	T/G	T	G	G	G	G	G	G	G
Tdurum_contig25513_123	5BL	B	280.68	A/G	A	G	G	G	G	G	G	G
Tdurum_contig25513_195	5BL	B	280.68	T/C	T	C	C	C	C	C	C	C
BobWhite_c48435_165	5BL	B	280.68	T/C	C	T	T	T	T	T	T	T
**RAC875_c62_1546**	**1DS**	**D**	**108.87**	**A/G**	**A**	**G**	**A**	**A**	**A**	**A**	**G**	**G**
BobWhite_c5419_643	1DS	D	108.87	A/G	A	G	A	G	A	G	A	A
**SNP**	**Chr.**	**Genome**	**Position (cM)**	**Alleles**	**Parents**		**Contrasting F_3_ Lines of Cross Bobur*UZ-11CWA08**					
							**P2G076**	**P2G243**		**P2G027**	**P2G162**	
					**UZ-11CWA08**	**Bobur**	**Tolerant**	**Tolerant**		**Sensitive**	**Sensitive**	
CAP7_c4879_249	1AL	A	313.85	A/C	A	C	A	A		C	A	
RAC875_c38018_278	2AL	A	110.13	T/C	T	C	C	C		C	T	
Excalibur_c20439_825	2AL	A	497.75	T/C	C	T	C	T		T	C	
Excalibur_c91176_326	2AL	A	502.77	A/G	A	G	A	G		A	G	
IAAV7086	2AL	A	544.94	A/G	G	A	G	A		G	A	
RFL_Contig5153_958	3AL	A	555.33	A/G	G	A	A	A		G	A	
Tdurum_contig33628_129	4AS	A	147.89	T/C	C	T	C	T		C	T	
Tdurum_contig33628_85	4AS	A	147.89	A/G	A	G	A	G		A	G	
wsnp_Ex_c43009_49439922	7AL	A	103.7	T/C	T	C	T	T		T	T	
BS00035083_51	7AL	A	103.7	T/C	T	C	T	C		T	T	
D_contig25392_201	1BS	B	195.12	A/G	G	A	G	G		G	A	
BobWhite_c11044_322	1BS	B	266.71	T/C	T	C	C	T		C	C	
Excalibur_c65341_303	2BS	B	365.88	A/G	G	A	A	A		A	A	
Ex_c16948_754	2BS	B	367.4	A/G	A	G	G	G		A	G	
BobWhite_c5756_532	2BS	B	583.38	A/C	A	C	C	C		C	C	
BS00032003_51	5BL	B	1.33	T/C	T	C	C	C		T	C	
Tdurum_contig25513_123	5BL	B	280.68	A/G	A	G	G	G		G	G	
**BobWhite_c48435_165**	**5BL**	**B**	**280.68**	**T/C**	**C**	**T**	**C**	**C**		**T**	**T**	
Excalibur_rep_c67190_638	7BS	B	228.36	T/G	T	G	G	G		G	G	
**BS00087086_51**	**1DS**	**D**	**108.87**	**T/C**	**T**	**C**	**T**	**T**		**C**	**C**	
**BS00002178_51**	**1DS**	**D**	**108.87**	**A/G**	**G**	**A**	**G**	**G**		**A**	**A**	

**Table 3 ijms-23-13745-t003:** Colocation of SNP clusters with QTL/genes.

Associated ST Traits	SNP	Contrasting F_3_ Lines	Chr.	QTL	R^2^ (%)	Position (bp)	Position (CM)
**ST_DRW**	BS00002178_51	Bobur*UZ-11CWA08	1DS	Q-1DS.1	≥13.33	33,712,262..33,712,362	108.87
**ST_DRW**	RAC875_c62_1546	Bobur*Altay2000	1DS	Q-1DS.2	≥13.33	32,543,884..32,543,984	108.87
**ST_DRW**	BS00087086_51	Bobur*UZ-11CWA08	1DS	Q-1DS.3	≥13.33	34,619,721..34,619,821	108.87
**ST_DRW**	Ex_c16948_754	Bobur*Altay2000	2BS	Q-2BS.1	≥12.69	699,826,968..699,827,068	367.4
**ST_DRW**	BobWhite_c48435_165	Bobur*UZ-11CWA08	5BL	Q-5BL.1	≥24.20	546,827,468..546,827,565	280.68

Note: ST_DRW Salt Tolerance Dry Root Weight.

## Data Availability

The data presented in this study are available on request from the corresponding author.

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
