# Peer review of "Validation of a QTL on Chromosome 1DS Showing a Major Effect on Salt Tolerance in Winter Wheat"

_ijms, 2022, doi:10.3390/ijms232213745_

Round 1

Reviewer 1 Report

Manuscript entitled “Validation of a QTL on Chromosome 1DS Showing a Major Effect on Salt Tolerance in Winter Wheat" submitted to International Journal of Molecular Sciences” is well written and the results are presented in a logical and coherent manner. The manuscript addresses the problem of salt stress and wheat sensitivity to this factor, which may have a negative impact on the yield. Moreover, the proposed genetic and molecular parameters that may be related to salt tolerance in plants and, in the longer term, may be used in the development of plants resistant to this stress. The aim of the work is clear and precise and the described methodology is correctly selected and varied. Although the manuscript is well-edited and developed , however, small improvements should be introduced that will improve its quality:

1.       In the Introduction part, the novelty of the study should be emphasized more;

2.       Please use citations in accordance with the journal's requirements;

3.       The quality of figures in manuscript and supplementary material should be improved.

Author Response

Dear Reviewer 1,                                                    

RE:   Submission of Revised Manuscript (#1964779)

Herewith we submit the revised version of manuscript Ref. #1964779 entitled “Validation of a QTL on Chromosome 1DS Showing a Major Effect on Salt Tolerance in Winter Wheat". The authors appreciate your time and efforts for handling this manuscript and for providing insightful comments and suggestions to improve its quality. We have revised the manuscript as per the reviewer’s suggestions. The point-by-point responses are provided below.

Response to Comments from Reviewer 1

Suggestions to Author/s

Manuscript entitled “Validation of a QTL on Chromosome 1DS Showing a Major Effect on Salt Tolerance in Winter Wheat" submitted to International Journal of Molecular Sciences” is well written and the results are presented in a logical and coherent manner. The manuscript addresses the problem of salt stress and wheat sensitivity to this factor, which may have a negative impact on the yield. Moreover, the proposed genetic and molecular parameters that may be related to salt tolerance in plants and, in the longer term, may be used in the development of plants resistant to this stress. The aim of the work is clear and precise and the described methodology is correctly selected and varied.

Response: Thank you for the appreciation of our work and manuscript.

Although the manuscript is well-edited and developed, however, small improvements should be introduced that will improve its quality:

  1. In the Introduction part, the novelty of the study should be emphasized more;

Response: We improved the introduction, included new findings and emphasized the novelty. The improvement addressed in lines [40-51], [83-106] and [114-118].

  1. Please use citations in accordance with the journal's requirements;

Response: corrected.

  1. The quality of figures in manuscript and supplementary material should be improved.

Response: Done using JPEG style for figures.

Reviewer 2 Report

The authors describe Validation of a QTL on Chromosome 1DS Showing a Major Effect on Salt Tolerance in Winter Wheat. In the descriptive section of the material, I recommend that the authors read and supplement the manuscript with a discussion with research about more chemical, physical or energetic aspects of biomass. I recommend reading the article: https://doi.org/10.3390/en14113270

In these articles, you can also find information about correlation using ANOVA with the Duncan test. Similar studies, in particular the division into homogeneous groups, were also missing in the article. You can find information about a fractional breakdown or CHONS analysis as well as correlation using ANOVA with the Duncan test. It will certainly enrich the manuscript.

Author Response

Dear Reviewer 2,                                                    

RE:   Submission of Revised Manuscript (#1964779)

Herewith we submit the revised version of manuscript Ref. #1964779 entitled “Validation of a QTL on Chromosome 1DS Showing a Major Effect on Salt Tolerance in Winter Wheat". The authors appreciate your time and efforts for handling this manuscript and for providing insightful comments and suggestions to improve its quality. We have revised the manuscript as per the reviewer’s suggestions. The point-by-point responses are provided below.

Response to comments from reviewer 2

Suggestions to Author/s

The authors describe Validation of a QTL on Chromosome 1DS Showing a Major Effect on Salt Tolerance in Winter Wheat. In the descriptive section of the material, I recommend that the authors read and supplement the manuscript with a discussion with research about more chemical, physical or energetic aspects of biomass. I recommend reading the article: https://doi.org/10.3390/en14113270

Response: Thank for your suggestion and for the reference. We have improved the introduction and discussion by including the information and ideas from the forwarded and several related articles.Whereas in the introduction the improvements were addressed in lines [40-51], [83-106] and [114-118] while in the discussion the improvements were addressed in lines [395-408], [441-443], [449-453], [475-476], [482-485], in the materials and methods the improvements were addressed in lines [587-588], [694-695] and in the conclusion part the improvements were addressed in lines [711-724].

In these articles, you can also find information about correlation using ANOVA with the Duncan test. Similar studies, in particular the division into homogeneous groups, were also missing in the article. You can find information about a fractional breakdown or CHONS analysis as well as correlation using ANOVA with the Duncan test. It will certainly enrich the manuscript.

Response: We recalculated the statistical analysis and we added in the revised text the Duncan test for comparing the means. The results are presented in all figures.
